# Magnetic Nanomaterials Mediate Electromagnetic Stimulations of Nerves for Applications in Stem Cell and Cancer Treatments

**DOI:** 10.3390/jfb14020058

**Published:** 2023-01-20

**Authors:** Lei Wang, Yefan Duan, Shujie Lu, Jianfei Sun

**Affiliations:** Jiangsu Key Laboratory for Biomaterials and Devices, School of Biological Science and Medical Engineering, Southeast University, Nanjing 210096, China

**Keywords:** cancer, regenerative medicine, electromagnetic stimulation, magnetic nanomaterials

## Abstract

Although some progress has been made in the treatment of cancer, challenges remain. In recent years, advancements in nanotechnology and stem cell therapy have provided new approaches for use in regenerative medicine and cancer treatment. Among them, magnetic nanomaterials have attracted widespread attention in the field of regenerative medicine and cancer; this is because they have high levels of safety and low levels of invasibility, promote stem cell differentiation, and affect biological nerve signals. In contrast to pure magnetic stimulation, magnetic nanomaterials can act as amplifiers of an applied electromagnetic field in vivo, and by generating different effects (thermal, electrical, magnetic, mechanical, etc.), the corresponding ion channels are activated, thus enabling the modulation of neuronal activity with higher levels of precision and local modulation. In this review, first, we focused on the relationship between biological nerve signals and stem cell differentiation, and tumor development. In addition, the effects of magnetic nanomaterials on biological neural signals and the tumor environment were discussed. Finally, we introduced the application of magnetic-nanomaterial-mediated electromagnetic stimulation in regenerative medicine and its potential in the field of cancer therapy.

## 1. Introduction

As cancer remains incredibly difficult to treat, it continues to be one of the major causes of mortality worldwide [1,2,3]. In clinical practice, although conventional treatments, such as chemotherapy, radiotherapy, and surgery, can improve the survival rate of patients to a certain extent, inevitable drawbacks remain [4,5,6,7,8,9]. Therefore, the current traditional cancer treatment methods cannot meet clinical needs, and new cancer therapies urgently need to be developed. One of the most appealing areas of current biomedical engineering is regenerative medicine [10,11,12]. The improvement of cell activity is crucial for the development of regenerative medicine. For instance, the inherent healing capability of wounded or damaged tissues is insufficient for cell migration, proliferation, and differentiation. Tissue regeneration that is “patient-friendly” might be accomplished if injured cells’ inherent ability to mend themselves can be increased by using scientific technology. One of the best ways to increase cell activity is to employ biomaterials [13,14]. Nanomedicine is obtained by combining biomaterials with specific scales and surface properties with drug standards. Nanomedicine can overcome the disadvantages of the weak targeting of small-molecule drugs, low bioavailability, and the poor metabolic behavior of traditional dosage forms, and show their unique advantages in clinical application [15]. 

Due to the development of nanotechnology, biomaterials have been widely used in cancer treatment and regenerative medicine, such as hydrogels, magnetic nanoparticles (MNPs) [16,17], liposomes [18,19], and metal-organic frameworks [20,21], etc. Among them, MNPs have the benefits of being non-toxic, bio-compatible, injectable, and having a high degree of accumulation in tissues or organs. Additionally, nanostructure iron and its oxides display super paramagnetism or paramagnetism under certain conditions, which not only maintain good magnetic responsiveness, but can also quickly recover to the state of no magnetization after removing the magnetic field; the remanence is zero, which makes its in vivo application much safer. At the same time, nanoscale iron-based particles can easily be phagocytic be degraded by cells in vivo, and then be decomposed into iron ions, which are stored in intracellular iron pools to enable the body to synthesize hemoglobin; they are also reused in other ways. Additionally, no obvious toxic side effects occur when the dose is moderate [22]. The United States Food and Drug Administration (FDA) has specifically authorized the clinical use of magnetic iron oxide nanoparticles [23]. Despite showing broad potential in the biomedical field, magnetic nanomaterials (MNMs) still have room to grow. 

Meanwhile, electromagnetism is an active factor in biological processes, and it has a broad therapeutic application in regenerative medicine and cancer treatment [24,25]. Therapeutics that utilize electromagnetic wave heat and ultrasonic cavitation to eradicate cancer cells have been widely studied [26,27]. In addition, studies have shown that tumor cell proliferation and angiogenesis can be inhibited by extremely low frequency (ELF) [28], pulsed electromagnetic fields [29], and the sinusoidal electromagnetic field, which can also cause apoptosis and noninvasive necrosis, but have no effect on normal lymphocytes [30]. Notably, electromagnetic fields are also able to regulate the nervous system and cell growth via stimulation [31,32]. Strategies based on electromagnetic stimulation have been widely used in the treatment of neurological diseases (depression, obsessive compulsive disorder, Parkinson’s disease (PD), etc.), stem cell differentiation, bone tissue engineering, nerve regeneration, and other fields [33,34,35,36], but a viable strategy has yet to be found in the treatment of cancer.

In contrast to other reviews in related fields, in this paper, we focused on the stimulation of biological nerve signals to modulate the regeneration or tumors, and because tumors can also be considered as types of uncontrollable cell regeneration, they can, therefore, be uniformly viewed in conjunction with regeneration. More specifically, the stimulation of nerves as a regulatory tool can be made more precise and controllable by innovatively combining magnetic nanomaterials and their assembled structures with external electromagnetic fields (Figure 1).

## 2. Effect of Biological Neural Signals on Stem Cell Differentiation and Tumor Development

### 2.1. Stem Cell Differentiation

Stem cells have become a hot spot in basic research and clinical trials due to their extremely high level of biosafety, and they are used in many different fields, such as neurological diseases, cardiovascular diseases, diabetes, blood diseases, and cancer. Stem cell therapy is completely different from traditional treatment methods because it mainly enables immune reconstitution by regulating cytokines and repairing damaged cell tissues. However, the uncertainty of stem cell differentiation often limits its practical application. Therefore, the precise use of electromagnetic stimulation or the stimulation of neurotransmitter secretion to precisely control the direction of stem cell differentiation is important in the exploration of the relationship between the nervous system and stem cell differentiation. The factors that affect stem cell differentiation can be classified as endogenous and exogenous. Endogenous influencing factors include differential gene expression, different gene expression levels, and luxury genes, etc. For example, stem cells can express specific genes to differentiate into target cells, and different gene expression levels and differential gene expression will cooperate to induce stem cell differentiation [37]. Exogenous factors mainly include the in-duction of intercellular differentiation and the extracellular matrix. For example, in a liver model, stem cells differentiate into liver cells under the mediation of factors secreted by liver cells. In a myocardial infarction model, mesenchymal stem cells (MSCs) secrete exosomes (Exos), and mir-125b in the Exos acts on cardiomyocytes, downregulates the autophagy level of cardiomyocytes through the p53/bnip3 signaling pathway, reduces the rate of the autophagic cell death of cardiomyocytes, and thus improves cardiac function [38]. Moreover, previous studies have shown that the direction of stem cell differentiation is regulated by the nervous system, which receives signals from target cells and secretes neurotransmitters that control the direction of stem cell differentiation [39].

At present, MSCs have been applied in a wide range of clinical applications, including in treatments of spinal cord injury, osteoarthritis, diabetes, myocardial infarction, and Alzheimer’s disease (AD), based on their ability to differentiate into multiple cell types, including myocardial cells, nerve cells, stem cells, epithelial cells, lung cells, tendons, and cartilage cells, etc. Predominantly, human embryonic stem cells have the potential to differentiate into MSCs, which further accelerates the loss of pluripotency markers and increases the expression of MSC surface markers by inhibiting the IκB kinase (IKK)/nuclear factor kappa B (NF-κB) signaling pathway (Figure 1) [40]. Hu et al. studied the mechanism of sympathetic action on the differentiation of MSCs, indicating that the EP4/prostaglandin E2 (PGE2) axis in sensory nerves directly controls the commitment of MSCs by regulating the sympathetic activity [41]. According to Wang et al., leptin from the hypothalamus influences osteogenesis via stimulation from the sympathetic nervous system, and can prevent MSCs from forming bone by inhibiting the β2 receptor on osteoblasts [42]. Electrical stimulation can also promote the differentiation of bone marrow MSCs into nerve-like cells by influencing the flow of ions through the cell membrane, changing the membrane potential, and regulating the intracellular signal transduction pathway [43,44]. It has been demonstrated that the nervous system also controls other stem cell differentiation processes. Gamma-aminobutyric acid (GABA), a major inhibitory neurotransmitter in the central nervous system, can bind with GABA_A_ receptors or GABA_B_ receptors, in order to control hematopoietic stem cell and progenitor cell proliferation and function [45].

### 2.2. Tumor Development

In recent years, the incidence and mortality rate of cancer have continued to rise, which is a cause for global concern [46]. To solve this problem, on the one hand, a variety of effective new therapies for the treatment of cancer have been developed; on the other hand, the mechanisms of tumor growth and spread have also been intensely studied, with the most notable area of research being the function of the nervous system in tumor development, progression, and metastasis. For instance, taurolithocholic acid (TLCA) induces the growth of intrahepatic cholangiocarcinoma (CHC) cells via activating the epidermal growth factor receptor (EGFR)/extracellular regulated protein kinases (ERK1/2) and muscarinic cholinergic receptors [47]. Notably, Lolas and colleagues made the first attempt to model the neurobiological aspects of cancer development through a system of differential equations. The model confirms the experimental observations that a tumor is able to promote nerve formation/elongation around itself, and that high levels of nerve growth factor and axon guidance molecules are recorded in the presence of a tumor. Their results also reflect the observation that high stress levels (represented by higher norepinephrine release by sympathetic nerves) contribute to tumor development and spread, indicating a mutually beneficial relationship between tumor cells and neurons (Figure 2) [48]. In this section, we focus on the relationship between the nervous system and cancer, with regard to a deep understanding of the profound effect of the nervous system on various cancers, including prostate, gastric, liver, and pancreatic cancer. 

According to the results of previous studies, the sympathetic nervous system (SNS) becomes overactive during the development of hepatocellular carcinoma [49]. For instance, several studies showed that the increased expression of the recombinant acetylcholinesterase (ACHE) protein dramatically reduced the development and tumorigenicity of hepatocellular carcinoma (HCC) cells in in vitro and in vivo organisms [50,51,52]. According to studies by Huan et al., the SNS causes Kupffer cells (KCs) to become more active and sustain an inflammatory environment by activating their α1-adrenergic receptors (α1-Ars), further indicating the essential role of the SNS in HCC cells [53]. In addition, it is amazing that some neurotransmitters show anti-tumor properties, such as dopamine (DA). Zhang et al. found that moderate exercise leads to the secretion of DA, enhances the activity of the D2 receptor, and inhibits the epithelial mesenchymal transition in transplanted liver cancer cells, which is induced by transforming growth factor-β (TGF-β1) [54]. Liu et al. found that SNS/β-adrenergic receptors (β-ARs)/CCL2 stimulate the peripheral SNS and β-ARs signaling in tumor cells and tumor-infiltrated myeloid cells, which results in the inhibition of CCL2 production and the activation of anti-tumor immunity. This finding suggests a possible therapy for anti-tumor immunity based on interventions in the nervous system [55]. 

Breast cancer cells (BCCs) secrete neuroactive chemicals, such as neurotrophic and axon guidance molecules, to encourage the expansion and branching of nearby peripheral nerves. While the parasympathetic nerve and sensory nerve mostly have anti-tumor actions in the advancement of breast cancer, the sympathetic nerve promotes the growth of breast cancer [56]. Peripheral nerves might, thus, influence the growth of breast cancer by indirectly acting on immune cells, in addition to directly binding to the appropriate BCC receptors via the release of neurotransmitters. Zeng et al. reported that breast cancer metastases may be triggered by parts of the primary tumor cells and under the activity of nearby neurons to stimulate growth, and that glutamic acid stimulation in the N-methyl-D-aspartate receptor ligands glutamate (NMDAR) signaling pathway aids the invasive growth of neuroendocrine and pancreatic tumors; commonly, the neuron signaling pathway is associated with a poor prognosis [57]. In addition to the fact that breast cancer cells can camouflage themselves, using signaling pathways in the body to speed up the growth of cancer cells and the death rate, glioma cells were found to have similar characteristics; it was shown that nerve cells secrete neurotransmitters that are absorbed by cancer cells to instantly increase the intracellular calcium ion concentration, which causes a series of reactions, and finally promotes the transfer of cancer cells [58]. 

This suggests that the nervous system is an emerging key factor in the promotion of tumor growth, including the conduction of neurochemical transmitters and the formation of tumor-neural synapses between neurons and tumor cells [59]. These mechanisms of action suggest that existing neuromodulating drugs may serve as potential anti-cancer therapies. Furthermore, the surgical or chemical dissection of specific nervous systems has a positive effect in cancer treatment, such as the surgical or chemical destruction of the sympathetic nerve in the prostate to inhibit early tumor growth and the pharmacological inhibition of the parasympathetic nerve to inhibit tumor metastasis [60,61]. This provides a sufficient theoretical basis for the use of magnetic nanomaterial-mediated electromagnetic stimulation in cancer treatment. Furthermore, we also summarized the relationship between neuronal signals or pathways and various cancers, and these are listed in Table 1. 

## 3. Effects of MNMs on TME and Biological Neural Signals

The earliest known examples of iron-based nanomedicine date back to 1957, when iron-based microparticles were reported to be used in the thermal ablation of lymphoma in vivo [74]. Due to the improvements in its surface modification, biocompatibility, stability, and functionality, iron-based nanomedicine is employed for the diagnosis or treatment of clinical disorders [75,76]. The FDA has approved the use of magnetic iron oxide nanomaterials Resovist^®^ (Berlin, Germany) and Feridex^®^ (Berlin, Germany) for the magnetic resonance imaging of liver tumors. Feraheme^®^ (Delaware, Waltham, MA, USA) is approved as an intravenous iron supplement for the treatment of iron-deficient anemia, among other conditions [77,78,79]. Due to their excellent in vivo compatibility, the steady controllability of batch preparation, and their affordable manufacture, iron-based nanomedicines have been employed extensively in clinical settings. Therefore, here, we will briefly introduce the effects of MNMs on the TME and biological neural signals.

### 3.1. Tumor Microenvironment

The tumor immunological microenvironment plays a key role in controlling an organism’s immune response to a tumor [80,81]. Despite new evidence suggesting active immunity, agents such as tumor vaccines can promote the infiltration and activation of T cells, natural killer (NK) cells, and dendritic cells (DCs), and increase the intensity of the immune response. During the development of tumors, the body can create a tumor immunological microenvironment that negatively affects the strength of the immune response to the tumor and impedes the immune system’s lethal impact on the tumor, via several methods. For instance, immune suppression can lead to a significant infiltration of immunosuppressive cells into the tumor site, including myeloid-derived suppressor cells (MDSCs), M2 macrophages, tumor-associated macrophages, regulatory T cells, and so on [82].

Extensive use has been made of magnetic nanoparticles in magnetic resonance imaging (MRI), magnetic targeting, and magnet-responsive drug delivery due to their outstanding controllability and magnetic driving force. Additionally, magnetic nanoparticles have been shown to control the activation of anti-tumor immunity. By manipulating magnetic fields, Schneck et al. created a platform for reductionist T cell activation [83]. Paramagnetic nanoparticles decorated with diverse signaling molecules form the basis of this platform. Each of these signaling molecules serves a unique purpose in the activation of the associated signaling pathway. However, T cell activation often necessitates the synchronization of some signaling molecules, such as costimulatory and T cell-receptor-specific signals. To increase T cell activation for immunotherapy, these single-signal nanoparticles might bind to the appropriate receptors and then use a magnetic field to stimulate the aggregation of these surface-bonded antigens or stimuli. 

The adoptive transfer of NK cells is a component of NK cell immunotherapy, which is gaining more attention as a possible immunotherapy for the treatment of a variety of diseases. Importantly, in contrast to many competing cytotoxic T lymphocyte therapies, NK cells may specifically destroy tumor cells without prior exposure to tumor-specific antigens. However, the success of the therapy is frequently only moderate in solid tumors that have already developed. To generate significant numbers of functional NK cells for cancer treatment, the ex vivo activation of NK cells with exogenous cytokines is commonly required but ineffective. Another option for treatment is the local delivery of NK cells under image guidance. There are not enough non-invasive methods to keep track of NK cells. To improve the therapeutic effectiveness of NK cells for solid tumors, Sim et al. created nanocomplexes (HAPF) made of therapeutically relevant materials (hyaluronic acid, protamine, and ferumoxytol) to enable the magneto-activation and MRI visibility of NK cells (Figure 3) [84]. In NK cells, an enhanced self-assembled HAPF nanocomplex was successfully bound and tagged. An exogenic AC magnetic field application could activate HAPF-labeled NK cells to improve their innate cytolytic capability. Actin filaments and NKG2D receptors, which are unique to NK cells, were activated. Then, as a result of the elevation of the NK cell activation markers Perf/Gzmb and NKG2D activation receptors, the cytolytic potential was enhanced. Additionally, MRI-visible HAPF-NK cells allowed for NK cells to be monitored after transcatheter hepatic intra-arterial local NK cell administration. In an HCC rat model, it was demonstrated that locally injected HAPF-NK cells, activated by a magnetic field, have potential therapeutic effects by slowing the development of solid tumors. 

Furthermore, Saeid et al. showed that ferumoxytol nanoparticles limit tumor growth through indirect effects on the TME: monocytes are attracted to malignant tumors by chemotactic cytokines and are normally polarized to anti-inflammatory M2 phenotypes [85]. Superparamagnetic iron oxide nanoparticles (SIONPs) have been demonstrated to shift the phenotype of M2 macrophages toward the high CD86+, tumor necrosis factor (TNF)-positive M1 macrophage subtype, according to earlier in vitro studies. A subsequent autocrine feedback loop that maintains the generation of TNF and nitric oxide (NO) can be created by the cancer cells’ continued M1 polarization, as a result of apoptosis. Therefore, the regulation of the tumor-immune microenvironment, mediated by magnetic nanoparticles, can achieve accurate and effective immune cell activation in development. 

### 3.2. Biological Neural Signal

By sending electrical, optical, chemical, auditory, or magnetic stimuli to specific neural tissue, a process known as neuromodulation may be used to alter neuronal activity [86]. This technique has given scientists important tools to both study how the brain works and to control the activity of damaged neural circuits to slow down the course of illnesses. The application of neuromodulation in neuroscience research has resulted in a plethora of findings regarding functional connectivity in brain networks. Furthermore, neuromodulation technologies, capable of enhancing, restoring, and replacing motor, sensory, and cognitive skills, have been used to develop therapeutic routes and devices for the treatment of neuropsychiatric disorders.

Through active research efforts, the development of nanotechnology has recently revolutionized neuromodulation techniques [87]. On the one hand, the adaptable nano-science toolbox promoted neuromodulation techniques that were previously associated with huge devices toward shrunk devices with soft mechanics, closely packed components, and long-lasting performance. The neurological issue may be seamlessly integrated with these nanoscale instruments due to their enhanced spatial resolution and precise targeting capabilities [88]. However, some of the drawbacks of conventional macroscale neuromodulation techniques may be overcome using nanomaterials with advantageous physical and chemical characteristics. For example, nanomaterials and nanoscale devices can add the advantage of a high spatial and temporal resolution to modulated models with a high penetration depth, thus enabling new grafted forms of neuromodulated models.

In addition, magnetic nanoparticles represent a significant aspect of magnetic neuromodulation development [89,90]. Deep brain stimulation is possible with the use of superparamagnetic nanoparticles, which can be delivered to the brain and controlled remotely. Additionally, by regulating certain ion channels, force-generating or heat-dissipating super-paramagnetic nanoparticles can be employed for wireless neuromodulation. Specific cells’ ion channels are targeted by taking advantage of their inherent functionality or via genetic modification. Magnetic nanoparticles’ magnetic forces activate mechanosensitive channels, such as TREK1 and Piezo1, and magnetic nanoparticles that produce heat in response to an external alternating magnetic field can activate heat-sensitive ion channels, such as TRPV1. The fact that there is no discernible change in the neuronal density or glial response between the stimulated and unstimulated patients is significant because it shows that little to no tissue damage is caused by the magnetic nanoparticles’ briefly dispersed heat. For example, Huang et al. were the first to use superparamagnetic nanoparticles with a changing magnetic field to generate heat and activate the temperature-sensitive TRPV1 channel, leading to the activity of some anesthetized worms. 

It should be noted that magnetic-nanomaterial-based neuromodulation methods only require the implementation of a low magnetic field intensity, compared to repetitive transcranial magnetic stimulation (rTMS), but both modulation methods mainly work in the 0–20 Hz region [91,92]. The International Commission on Non-Ionizing Radiation Protection (ICNIRP) suggested that the exposure limit is frequency-dependent [93]. The upper limit for continuous public exposure in the 1–8 Hz frequency band is around 400 Oe. In addition, tissue heating and other negative side effects may result from strong AC magnetic fields. Therefore, in comparison to high-strength rTMS, the use of low-strength magnetic fields in nanoparticle-mediated treatments is advantageous. 

For instance, by stimulating the signaling pathway for the mitogen-activated protein kinase, Fe_3_O_4_ nanoparticles can also promote neurite development. The ability to use nanoparticles to improve the mechanical characteristics of the nerve guiding conduit (NGC) has also been demonstrated in earlier research, which is even more significant. Chen et al. created a multilayered composite NGC (ML-NGC), loaded with melatonin (MLT) and Fe_3_O_4_ by electrospinning it (Figure 4) [94]. With MLT minimizing oxidative damage and Fe_3_O_4_ promoting neurite renewal, this three-layer scaffold is thought to offer enough mechanical strength for neurite sprouting; it will, therefore, establish the ideal milieu for nerve regeneration. 

## 4. MNMs for Regenerative Medicine and Cancer Therapy through Electromagnetic Stimulations

Due to their distinctive physical characteristics, chemical properties, and biological impacts, nanomedicine based on MNMs should receive special attention and support as an important aspect of nanomedicine. Therefore, we briefly outline some biomedical applications of MNMs in this section.

### 4.1. Stem Cells

#### 4.1.1. Stem Cell Regeneration and Differentiation

Stem cells have a great capacity for self-renewal and a great multilineage differentiation potential; thus, they are regarded as possible therapeutic vehicles in regenerative medicine [95,96,97]. However, there are many limits, such as ethical and legal issues, within regenerative therapies that focus on stem cells; further research must be carried out regarding the safety and efficiency of artificial organs [98]. In cell therapies, stem cells are directly applied to damaged tissues, but currently, cell regulation cannot be operated accurately enough [99]. Seeding cells expanded in vitro on appropriate scaffolds is another suggested approach to tissue reconstruction. Previous studies have shown that stem cells respond to physiological forces and then differentiate to specific cell types via sonic vibrations, electromagnetic fields, magnetic forces, and stress relaxation [100,101]. The magnetic field not only affects the activity of cells and contributes to nerve regeneration, but also controls MNP-based nanomotors’ motion due to their mechanical and easily-controlled properties. On some occasions, MNPs are wrapped in materials such as hydrogel and silk fibroin to influence the target tissue. As a result, MNPs are ideal cell microcarriers that are suitable for cell seeding, cultivation, and further delivery under the control of a magnetic field for stem cell regeneration and differentiation [102,103]. 

According to Semeano et al., the migration and neuronal maturation of murine embryonic stem cells and human-induced pluripotent stem cells are made easier by the presence of 0.5% MNPs in collagen-based coatings [104]. Additionally, the application of an external magnetic field of 0.4 Tesla, perpendicular to the cell culture plane, stimulates proliferation and controls the pluripotent stem cells’ (PSCs’) decisions regarding their fate; this depends on the origin of the stem cells and their stage of maturation. The modification of ionic homeostasis and the expression of proteins involved in cytostructural, liposomal, and cell cycle checkpoint functions are the primary mechanisms by which electromagnetic stimuli affect neural lineage specificity and proliferation, according to mechanistic research. The results of previous research verify the feasibility of using MNPs in targeted stem cell transportation and transplantation in in vitro and in vivo physiological fluidic environments, and the use of appropriate magnetic stimuli leads to a highly efficient and controllable motion [105].

Magnetic nanoparticles can produce mechanical effects in gradient or rotating magnetic fields. The “nano-magnetic force” can be used to regulate cell function and fate, such as damage to the cell membrane system or cytoskeleton, resulting in the apoptosis or necrosis of cells; it can also be used to promote stem cell differentiation and the formation of functional tissues. For instance, a magnetic force bioreactor (MFB) was created by Hu et al. and uses magnetic micro- and nano-particles to target cell surface receptors to apply highly tailored local stresses to cells at a pico-newton level (Figure 5) [106]. The MFB system consists of horizontal arrays of NdFeB magnets, onto which cell culture plates can be situated. The frequency and amplitude of the oscillations of the array are controlled via a computerized stepper motor system. The field strength produced by the magnetic arrays of the MFB in the vicinity of the cells is in the region of 60–120 mT, with a field gradient of 3.3–11.0 Tm^−1^. The findings demonstrated that human bone-marrow-derived mesenchymal stem cells (hMSCs) greatly improved osteogenic differentiation via magneto-mechanical stimulation, mediated by MNPs.

#### 4.1.2. Bone Tissue Engineering

Orthopedic disorders, including infections, cancers, and bone loss, continue to represent problems in global public health [107,108,109]. As populations get older, this problem assumes additional significance because degenerative disorders, such as osteoporosis, can increase the likelihood of bone fractures and make it more challenging to mend broken bones. Clinical research regarding bone repair scaffolds focuses on improving cell adhesion, mechanical strength, and the active stimulation of local ecological niches. However, using molecular modifications to control surface properties uniformly and compactly remains a challenging task. It has also been discovered that magnetic fields significantly improve fracture healing and bone development [110]. Among us, magnetic fields include static magnetic fields (SMFs), pulsed electromagnetic fields, rotating magnetic fields and alternating electromagnetic fields. SMFs and pulsed electromagnetic fields were the two most studied types of magnetic fields. The mechanism of SMFs, used to promote osteogenesis, was likely because the cell membrane possessed diamagnetic properties, and the exposure to SMFs served to modify the membrane flux [111,112,113]. In addition, the extracellular matrix proteins had diamagnetic properties, and their structures and orientations could be affected by the SMFs. Notably, after implantation, magnetic effects can also be wirelessly connected by an outside field [114]. Therefore, by combining various functionalities, magnetic scaffolds can play a distinctive role in clinical bone restoration. It is easy to modify SIONPs onto scaffolds and magnetize them with outside magnetic fields. Additionally, SIONPs have properties in common with biomacromolecules, such as cell attachment sites, and because of their nanoscale and biosafety, they are highly advantageous for use in magnetic implants. For example, Chen et al. reported the fabrication of magnetic poly (lactic-co-glycolic acid)/polycaprolactone (PLGA/PCL) scaffolds via electrospinning and the layer-by-layer assembly of SIONPs [115]. The same technique was used to create PLGA/PCL scaffolds, which were constructed with gold nanoparticles for comparison. The findings showed that the nanoparticle film that formed on the surface significantly improved the scaffold’s hydrophilicity and increased its elastic modulus, which, in turn, boosted the osteogenesis of adipose-derived stem cells (ADSCs). Furthermore, it was shown that the key component favoring osteogenic differentiation was the magnetic characteristics of SIONPs magnetized by external magnetic fields, which explained why the magnetic scaffolds had a better osteogenic capability than the gold-nanoparticle-assembled scaffold. These results underline the relevance of magnetic nanoparticles as a bioactive interface between cells and scaffolds, and support the design of biomaterials to improve the efficacy of tissue engineering and regenerative medicine. 

With or without SMFs, magnetic nanoparticles can stimulate angiogenesis and osteogenesis. According to earlier research, Exos, produced by bone marrow mesenchymal stem cells (BMSCs), have therapeutic effects that are like those of BMSCs in treating bone regeneration; they deliver Exos and seldom trigger strong immune reactions. The most significant finding was that BMSC-Exos, cultivated under magnetic conditions, also improved wound healing. Therefore, Wu et al. demonstrated that Exos derived from BMSCs, pretreated with magnetic fields or low-dose magnetic nanoparticles, can promote osteogenesis and angiogenesis during bone regeneration (Figure 6) [116]. Among them, it was shown in both in vivo and in vitro experiments that the BMSC-Fe_3_O_4_-SMF-Exos, produced after the combined pretreatment of magnetic fields and magnetic nanoparticles with mesenchymal stem cells, had the most significant osteogenesis and angiogenesis effects. This promoting effect is related to the high abundance of miR-1260a in BMSC-Fe_3_O_4_-SMF-Exos. In addition, some other experimental results have shown that a plasmid that overexpresses HDAC7 in BMSC and a plasmid that overexpresses COL4A_2_ in HUVEC could abolish the promoting effect of exosomal miR-1260a mimics on osteogenesis and angiogenesis. The overexpression of HDAC7 rescued osteogenic activity, the overexpression of COL4A_2_ rescued angiogenic activity, and both were enhanced by miR-1260a mimics. They concluded that exosomal miR-1260a, derived from BMSC-Fe_3_O_4_-SMF-Exos, promoted osteogenesis by targeting HDAC7 and that it promoted angiogenesis by targeting COL4A_2_.

#### 4.1.3. Neurological Diseases Treatment

Neurodegenerative diseases, such as AD, PD, and Huntington’s disease, represent a current medical challenge. Neurodegenerative diseases are characterized by the degeneration or death of neurons, leading to nervous system dysfunction that affects motor and memory abilities. Currently, stem cell therapy represents a successful method for the treatment of neurodegenerative diseases. This is because inducing stem cells to differentiate into neurons and repairing or replacing damaged and missing neurons can promote the recovery of lost function. The proliferation and differentiation of stem cells, early neuronal development, and early neuronal migration have all been shown to be critically dependent on the electrical activity of a cell. Therefore, through electrical stimulation, stem cell differentiation can be regulated to achieve the directed growth of neurites. For example, Han et al. combined micro–nano processing technology to achieve the accurate and controlled preparation of graphene scaffold materials, and conducted an in-depth study regarding the influence of scaffold dimension, topology, size, and other factors on neural stem cell (NSC) differentiation behavior [117]. A graphene scaffold is designed as a circular ring to form a closed loop. Using the principle of electromagnetic induction, after an alternating current is applied to a conductive coil, the graphene ring can generate an induced current in the loop due to the change in the magnetic field. As the cell is stimulated by external electricity, the charge distribution around the cell membrane can be changed, and the cell can generate an action potential; this affects the cell behavior and achieves the purpose of regulating the behavior of stem cells via radio stimulation. Notably, previous studies have shown that NSC differentiation can also be stimulated by extremely low-frequency electromagnetic fields (ELFEFs) that affect several biological parameters, such as the intracellular calcium level [118,119]. ELFEFs upregulate the Ca^2+^ channels and increase the Ca^2+^ influx, which both induce the signaling cascade associated with the promoter of the specific basic helix–loop–helix that controls NSC differentiation. For examples, Choi and co-workers explore the first step in neural differentiation by MNPs, in combination with physical guidance using ELFEFs [120]. In this study, synthesized PEG-phospholipid encapsulated magnetite (Fe_3_O_4_) nanoparticles are used on human bone marrow-derived mesenchymal stem cells to improve their intracellular uptake. The PEGylated nanoparticles were exposed to the cells under 50 Hz of ELFEFs to improve neural differentiation.

In addition, one of the most important factors contributing to the pathophysiology of PD is oxidative stress. In addition to other biological factors, the disorder is linked to unusually high amounts of reactive oxygen species (ROS) produced by cells, changes to the mitochondrial electron transport chain, and the accumulation of iron deposits in the substantia nigra pars compacta. Due to this, recently, antioxidant-rich nanomaterials have been produced for use in PD therapy. In a recent study, Umarao et al. physically implanted SIONPs into the striatum before exposing rats to magnetic fields to study the neuroprotective potential of this approach in a 6-OHDA rat PD model [121]. In this method, the Fe^2+^ and Fe^3+^ ions on the surface of SIONPs function as free radical scavengers, giving the NPs antioxidant capability. Additionally, being exposed to an electromagnetic field enhanced the antioxidant action of NPs by modifying the likelihood that newly created pairs of radicals will recombine, changing the number of free radicals present and the degree of oxidative stress [122,123,124]. Finally, some applications and working mechanisms of MNMs in regenerative medicine are summarized in Table 2.

### 4.2. Potential of MNM—Mediated Electromagnetic Stimulation of Nerves in Cancer Treatment

Immature myeloid cells (IMCs), produced in bone marrow in healthy individuals, can differentiate into mature granulocytes, macrophages, or DCs. However, in a pathological setting, such as in cancer, different infectious diseases, and some autoimmune disorders, a partial block in the differentiation of IMCs into mature myeloid cells leads to an increase in this immature population. NO and ROS are produced in greater quantities when this IMC is activated under pathogenic circumstances, upregulating the expression of immune suppressive factors, such as iNOS and Arg-1 [125,126]. Classic immunosuppressive cytokines, including IL-10 or TGF-β, are also produced, which leads to the inhibition of T-cell responses in the TME while an IMC population is growing. IMCs have been shown to have non-immunological activities in cancer, including the promotion of angiogenesis, tumor cell invasion, and metastasis, in addition to their suppressive effects on immune responses. These cells have been collectively dubbed MDSCs [127,128]. The proliferation and expansion of myeloid progenitors that give birth to MDSCs are decreased by the overexpression of trefoil factor 2 (TFF2), an anti-inflammatory peptide released by the spleen [129]. Then, it was shown that overexpressing TFF2 significantly inhibited tumor development. According to an experimental mouse investigation, the efferent vagal pathway is crucial for TFF2 production in splenic T-cells and, in turn, for the prevention of MDSC proliferation in colorectal cancer [129]. The electrical stimulation of the vagus nerve (VNS) elevated the TFF2 expression. In contrast, the splenic TFF2 response was stopped in mice with bilateral subdiaphragmatic vagotomy, leading to an increase in MDSCs and colon carcinogenesis. It is important to note that the addition of isoproterenol, a substance that mimics post-ganglionic vagal stimulation, allowed for a partial recapitulation of this circuit in vitro. 

In addition, many similar cases show the role of nerves in tumor regulation, which may further expand the application of electromagnetic stimulation for tumor therapy. For instance, Erin et al. showed that CNI-1493, commonly known as Semapimod, stimulating the VNS, reduced the initial tumor development in a mouse breast cancer model [130]. Additionally, compared to the control group, the Semapimod-treated mice exhibited considerably fewer macroscopic liver and lung metastases. Semapimod may thereby prevent the metastasis of breast cancer by stimulating the VNS. Overall, we anticipate that the electromagnetic stimulation of the VNS will be less harmful than pharmaceutical stimulation, in terms of side effects in preventing tumor development and spread. 

Compared with the electrical stimulation of the VNS, magnetic stimulation shows outstanding performance in terms of penetration depth and non-invasiveness, while magnetic-nanoparticle-mediated vagus nerve electromagnetic stimulation rectifies the low accuracy in traditional magnetic stimulation. Although drugs have now been used in some studies to stimulate nerves to treat cancer, the potential side effects and addictive nature of drugs are matters for discussion. In fact, for nerve stimulation, it is not always necessary to use drugs, and intervention using external electromagnetic fields may achieve the same effect as drug stimulation or other stimulation. At present, our group has made some progress in the application of magnetic nanoparticle-mediated electromagnetic stimulation in the treatment of central nervous system diseases. However, this method has not been widely used in vagus nerve stimulation. Therefore, we boldly predict that the vagal nerve electromagnetic stimulation technology, based on magnetic nanoparticles, will be brilliant for use in the field of cancer treatment.

## 5. Conclusions and Prospects

In the past few years, due to advancements in nanotechnology and brain science, MNMs and biological nerve signals have been extensively studied in terms of regenerative medicine and cancer therapy. In addition, emerging nanotechnology also enables neuromodulation in less invasive ways, with improved biological interfaces, deeper penetration, and greater spatiotemporal precision. Based on this, in this paper, we reviewed the effects of biological nerve signals on stem cell differentiation and TME, as well as the effects of MNMs on the TME and nerve signals. Combined with these two points, the application of MNMs in regenerative medicine through electromagnetic stimulation (such as stem cell regeneration and differentiation, bone tissue engineering, neurodegenerative diseases, etc.), and their potential in cancer treatment, are discussed. However, challenging issues remain with regard to this therapeutic method. For example, changes in the activity of the affected neuronal cells can be detected by smaller or non-invasive devices, or methods during magnetic-nanomaterial-mediated electromagnetic stimulation, thus responding to the stimulation effect in real-time. In addition, we must consider how better MNMs with more controllable diameters that cross the blood–brain barrier more easily and target tumor tissues via intravenous injection, which thus avoid the invasiveness associated with localization or in situ injection, can be prepared. We intend to address these questions in the future.

Although some issues still need to be addressed, we believe that the dynamic and continuous development of magnetic nanotechnology and brain science will provide a full and rich cross-section of ideas in regenerative medicine and oncology, and promote further clinical applications of MNMs in both fields; this will address the present-day reliance on invasive electrical stimulation and drug therapy for neurological diseases, and the problems of metastasis, recurrence, drug resistance, and side effects that accompany cancer treatment.

## Data Availability

Not applicable.

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
