# Peer review of "Magnetic Nanomaterials Mediate Electromagnetic Stimulations of Nerves for Applications in Stem Cell and Cancer Treatments"

_jfb, 2023, doi:10.3390/jfb14020058_

Round 1

Reviewer 1 Report

The review paper prepared by Lei Wang and collegues is devoted to the use of magnetic nanomaterials for regenerative medicine and the treatment of cancer using electromagnetic stimulation. Although this topic is very interesting and important, I do not recommend this review for publication. There are many research papers and reviews devoted to this research topic, which this review cannot compete with. Unfortunately, this review is not enogth informative, physical mechanisms of action and properties of materials are weakly discussed, contains factual errors and omits many important works in this direction. 

Additional comments:

1) In the abstract, the authors compare magnetic stimulation and electromagnetic stimulation mediated by magnetic nanomaterials but do not give a clear distinction between these mechanisms.

2) The authors should emphasize the novelty of this review  compared with lots of published literature.

3) The authors write "In this method, the Fe2+ and Fe3+ atoms on the surface of SIONPs function as free radical scavengers, giving the NPs 447 antioxidant capability." but, in oppposte, iron oxide 2+/3+ is known for its reactive ability to produce ROS follow Fenton-like reactions. 

4) Many of the discussed works refer only to materials without electromagnetic simulation or to the use of simulation without magnetic nanomaterials.

5) Abbreviations are not always introduced appropriately, e.g., introduced and not used further (e.g., MOF). 

Reviewer 2 Report

1. The numbers of figures in the text and in the legend should be corrected. For both figures 1 and 2, figure 1 is written.

2. Has permission been obtained for the copyrighted figures?

3. The manuscript needs to be corrected and checked for spelling and grammatical errors.

4. All "et al." should be written in italics.

5. Table 2 should be expanded and more columns should be added to explain more about the type of stem cell, the tests performed and the results of the studies.

6. The limitations and challenges of using electromagnetic simulations in regenerative medicine and cancer treatment should be written.

7. The future perspective of this method is suggested to be written.

Reviewer 3 Report

The review by Sun et al. mainly summarized the effects of biological nerve signals on stem cell differentiation and tumor microenvironment, and the effects of magnetic nanomaterials on tumor microenvironment and nerve signals. Generally, the review is well organized and written. However, the content of the review mainly focused on biological nerve signals on stem cell differentiation and tumor microenvironment, so I do not think the title of the review is appropriate. I strongly recommend the authors change the title before this review can be considered to be accepted for publication.

Some other comments as follows.

1. Figure 1. Magnetic nanomaterials mediated electromagnetic stimulation for regenerative medicine and cancer treatment. This should be deleted because it is not necessary and there is another Figure 1 exists.

2. The authors should check the abbreviations carefully. Some did not have full name and some did not appear at first time.

3.  And nanostructure iron or its oxides have super paramagnetism or paramagnetism under certain conditions, which not only maintains good magnetic responsiveness, but also can quickly recover to the state of no magnetization after removing the magnetic field, and the remanence is zero, which greatly enhances the safety of in vivo application. At the same time, nanoscale iron-based particles are easy to be phagocytic and degraded by cells in vivo, and then decomposed into iron ions, which are stored in intracellular iron pools for the body to synthesize hemoglobin and other reuse, and no obvious toxic side effects will occur when the dose is moderate (p.1-2).

Under certain circumstances, the nano-structured iron or its oxide exhibits superparamagnetism or paramagnetism, which not only maintains good magnetic responsive ness but also quickly returns to a state of no magnetization after removing the magnetic field, and the remanence is zero. This significantly increases the safety of in vivo applications. At the same time, when the dose is moderate, it is simple for cells in the body to phagocytose and degrade iron-based nanoparticles. These particles are then broken down into iron ions and stored in intracellular iron pools, where they can be used again by the body to synthesize hemoglobin and other substances without obvious toxic and side effects (p.6).

This two paragraphs are highly repetitive.

Round 2

Reviewer 1 Report

The authors have clarified the novelty of this review and, considering the positive feedback from the other reviewers, I recommend this manuscript after minor revisions. 

Figures 2 and 6: the textual description of these figures is not enougth full. In the case of Fig.6, the right panel (with HUVEC cells) of the figure is not explained in the text of manuscript.

Figure 3: please, increase the size of the image, the text is unreadable. 

Figure 4: please, remove the letter (A).

line 13: "bio-logical" -> "biological";  "nano-materials" -> "nanomaterials".

line 15:  "...(thermal, electrical, magnetic, force, etc.)" Please, change the word "force" to "elastic" or "mechanical".

line 354: please, remove "(EMF)", this abbreviation is not used hereafter. I suggest re-checking this point and additionally removing synonymous abbreviations, e.g. MNPs, NPs, and SPIONs, if the differences between the terms are not taken into account. 

line 451: "...Fe2+ and Fe3+ atoms" -> "Fe2+ and Fe3+ ions"; atoms are electrically neutral, ions have a charge.

lines 368-373: if possible, specify the parameters of the magnetic field created by this bioreactor.

One of perhaps the most interesting mechanisms of the effect of magnetic nanomaterials on stem cells is based on the magnetoelectric effect, which simplistically can be explained by the electrical polarization caused by mechanical stress in the piezoelectric matrix by magnetic inclusions in an alternating magnetic field. This approach allows to control the microenvironment of stem cells to induce controlled differentiation and proliferation (10.1016/j.mtbio.2021.100149) or, for example, to stimulate neuronal cells (doi.org/10.1002/adhm.202100695). It may be worthwhile to note the mechanism based on "magnetoelectric effect" somewhere in sections 4.1.2 Bone tissue engineering (e.g., doi.org/10.1021/acsami.7b19385; doi.org/10.3390/gels8100680) and 4.1.3 Neurological disease treatment (e.g., doi.org/10.1002/jcp.28040; doi.org/10.3390/nano11051154).
